# Characterization of Acid-Soluble Collagen from Food Processing By-Products of Snakehead Fish (*Channa striata*)

Thi Mong Thu Truong [1], Van Muoi Nguyen [2] 🆔, Thanh Truc Tran [2,*] 🆔 and Thi Minh Thuy Le [1,*]

1 Department of Fisheries Products Processing, Can Tho University, University Campus II, Can Tho City 94000, Vietnam; ttmthu@ctu.edu.vn
2 Department of Food Technology, College of Agriculture, Can Tho University, University Campus II, Can Tho City 94000, Vietnam; nvmuoi@ctu.edu.vn
* Correspondence: tttruc@ctu.edu.vn (T.T.T.); ltmthuy@ctu.edu.vn (T.M.T.L.); Tel.: +84-90-971-2070 (T.T.T.); +84-96-906-2679 (T.M.T.L.)

**Abstract:** The isolation of acid-soluble collagen (ASC) from by-products of snakehead fish (*Channa striata*), including skin and the mixture of skin and scale, has been investigated. The recovery yield of fish skin ASC (13.6%) was higher than ASC from fish skin and scale (12.09%). Both ASCs were identified as type I collagen and showed maximal solubility at pH 2. Collagen samples from the mixture of skin and scale had higher imino acid content (226 residues/1000 residues) and lower wavenumber in the amide I and amide III region (1642 and 1203 cm$^{-1}$, respectively) than the fish skin ASC (the imino acid content was 220 residues/1000 residues and the wavenumber in the amide I and amide III were 1663 and 1206 cm$^{-1}$, respectively. The difference scanning calorimeter (DSC) showed higher thermal stability in ASC from the mixture of skin and scale (T$_d$ of 35.78 °C) than fish skin ASC (34.21 °C). From the result, the denaturation temperature of ASC had a close relationship with the content of imino acid as well as with the degradation of α-helix in amide I and III. These results suggest that collagen could be obtained effectively from snakehead fish by-products and has potential as a realistic alternative to mammalian collagens.

**Keywords:** acid-soluble collagen; snakehead fish; fish skin; the mixture of skin and scale; denaturation temperature; FTIR

## 1. Introduction

The farming of snakehead fish, especially common snakeheads (*Channa striata*), a popular freshwater fish species, has rapidly spread in Vietnam [1]. Snakehead fish is a popular species and widely consumed in fresh and processed products in the Mekong Delta [2]. Snakehead production has developed rapidly in recent years; total production increased from 5300 tons to 40,000 tons during a 2002–2009 period [1]. The main processed products from snakehead fish in Vietnam contained dried snakehead fish and fish balls. The main steps to produce dried snakehead fish were, removed scales, fillet, remove viscera, and skin. For producing fish balls, the main steps were to fillet and separate the fish meat and the mixture of skin and scale. The by-products discarded during processing mainly consisted of head, skin, bone, fins, scales, and viscera, and accounted for 54% of the snakehead fish weight [3]. The huge amount of these by-products caused serious ecological issues. Furthermore, it was also becoming a potential material for the production of value-added products, especially for the extraction of collagen and gelatin [4].

Collagen has a huge application in food science for forming emulsions, edible films, or using as an agent of antimicrobial, antioxidant, and anti-hypertensive properties during food storage. Furthermore, collagen and gelatin protein hydrolysis could be used as a material for edible film production [5]. The utilization of fish by-products as a material for collagen extraction has been of interest in recent years [6] by the absence of disease transmission and dietary restrictions [7] when compared to the initial sources

of collagen extraction as bovine and porcine. Some previous research utilized different fish by-products for collagen extraction, such as the skin of tra catfish, clown knifefish, tilapia [8], golden fish [9], or from the scales of horse mackerel or flying fish [10]. However, the recovery yield of fish collagen is mainly dependent on the kind of fish by-products and fish species involved [11]. Furthermore, some important characteristics of collagen, such as the denaturation temperature, have a close correlation with fish habitat and imino acid content [10,12]. However, we find no information on the collagen extraction from snakehead fish by-products. Therefore, the aim of this study was to isolate acid-soluble collagen (ASC) from snakehead fish skin and the mixture of skin and scale to compare the extracted ASC characteristics and extraction yields between the representative materials for selecting a suitable collagen extraction material.

## 2. Material and Methods

### 2.1. Collected and Pretreated Skin and the Mixture of Skin and Scale

Skin and the mixture of skin and scale of snakehead fish were collected from a dried snakehead fish and fish ball company located in An Giang province, Vietnam. The main steps for preparing these by-products from whole snakehead fish were, (i) wash in chilled water, store under ice conditions, and transfer to the laboratory immediately; (ii) cut sample into small pieces and put into PE bags; and (iii) store at $-20\,°C$ until used.

### 2.2. Acid Soluble Collagen Extraction (ASC)

ASC was isolated according to the method of [10] with slight adjustments. The snakehead fish skin was soaked and stirred gently in NaOH 0.1 M for 6 h (changing solution every 3 h) at $4\,°C$ with a material/alkaline solution ratio of $1/8$ (*w/v*) to remove the non–collagenous proteins. The mixture of skin and scale were treated to demineralization in EDTA-2Na 0.8 M for 24 h (change solution every 12 h) at $4\,°C$ at a material/solution ratio of $1/8$ (*w/v*). After pretreated, the fish skin and the mixture of skin and scale were washed completely in cold distilled water until a neutral pH was obtained. The collagen extraction of skin and the mixture of skin and scale were conducted by using 10 volumes of acetic acid 0.5 M for 3 days at $4\,°C$. The solution obtained after extraction was continuously treated with these main steps: (i) centrifuging at $9000\times g$ for 20 min at $4\,°C$ to collect the supernatant; (ii) using NaCl for salt-out until obtaining a final concentration of 2.6 M in the presence of tris (hydroxymethyl) aminomethane 0.05 M at pH 7.0; (iii) centrifuging again at $9000\times g$ for 20 min at $4\,°C$ to collect the precipitate; (iv) dissolving in 0.5 M acetic acid, dialyzing and lyophilizing at $-20\,°C$ for 12 h; then (v) collecting the ASC by using a freeze-dryer.

### 2.3. Yield of Extracted ASC

Basing on the hydroxyproline content (Hyp) in extracted ASC and the crude material, the extraction yield of ASC was calculated following the described of [13] as below:

$$\text{Yield (\%)} = \frac{\text{Hyp content in extracted ASC (mg/L)} \times \text{Total extracted volume (L)}}{\text{Hyp content in crude material (mg/g)} \times \text{dry weight of crude material (g)}}$$

### 2.4. Viscosity of ASC

The collagen sample was dissolved completely in 0.1 M acetic acid to obtain a solution with a final concentration of 6.67%. The viscosity of the collagen solution was measured as described by [14] by using a digital viscometer (Brookfield DV, RVDV-11+CP, Middleboro, MA 02346, USA) with the speed of the spindle at 100 rpm.

### 2.5. The Solubility of Collagen at Different pHs and Various NaCl Concentrations

For determining the changing ASC solubility at different pH, a collagen sample at a final concentration of 3 mg/mL was prepared by dissolving in 0.1 M acetic acid and separated into 10 parts, the volume of each part was 8 mL. We adjusted pH of each part

by using either 6 M HCl or 6 M NaOH from 1 to 10 and then volume up to 10 mL by adding distilled water. Each 10 mL of collecting solution was stirred for 30 min at 4 °C and continuously centrifuged at 9000× *g* at 4 °C for 20 min. The resulting supernatant was collected, and protein content was checked using Lowry's method [15] with bovine serum albumin as a protein standard. The solubility of the supernatant at each pH was calculated by comparing it with the pH showing the highest solubility by the equation below:

$$\text{The solubility at each pH (\%)} = \frac{\text{The protein content at each pH}}{\text{The highest protein content}} \times 100$$

For testing the changing of ASC solubility at various NaCl concentrations, the collagen solution at the concentration of 6 mg/mL was obtained by dissolving collagen in 0.1 M acetic acid. Collagen solution was separated into 6 parts, with the volume of each part being 5 mL. Mixing each part with 5 mL of NaCl solution at 0.2, 0.4, 0.6, 0.8, 1, and 1.2 M while stirring for 30 min at 4 °C, and then centrifuging at 8000× *g* at 4 °C for 20 min. The resulting supernatant was collected and checked protein content as described as above. The supernatant obtained at each NaCl concentration was checked solubility and compared with the NaCl concentration showing the highest solubility.

### 2.6. Amino Acid Analysis

The amino acid composition of ASC from the snakehead fish by-products was analyzed according to the method of [10]. Approximately 0.1 g of ASC was measured into a media bottle for each analysis. Then we added 5 mL of oxidation solution into the bottle while continuously stirring and placed it into a refrigerator at 0 °C for 16 h. Subsequently, these ASC samples were combined with 0.84 g of sodium disulfite and 25 mL of hydrolysis solution. After that, the ASC samples were hydrolyzed by furnacing at 110 °C for 23 h. The hydrolytes were derivatized, using sample diluents for diluting, and compared with the analyzed standard of amino acids. was used for Amino acids analysis was conducted with 20 μL of the sample by using the Amino Acid Analyzer system (Biochrom 32+, Holliston, MA 01746, USA). The data were expressed in amino acid residues/1000 residues of amino acid contents.

### 2.7. The Molecular Weight of Collagen by SDS-PAGE

The protein profiles of ASC were accomplished referring to [16], as described by [8], with slight adjustments. ASC samples dissolved in sample buffer contained Tris-HCl 0.5 M, pH 6.8, including SDS 10% (*w/v*) and glycerol 20% (*v/v*) in the presence of mercaptoethanol 10% (*v/v*) at a ratio of collagen/sample buffer 1/2 (*v/v*); then they were loaded onto polyacrylamide gel 7.5% (Biorad, Hercules, CA 94547, USA) with protein molecular weight markers (Sigma Chemical Co., St. Louis, MO, USA). After being separated, gels were stained and destained. By comparing with protein markers, the molecular weight of protein bands was estimated.

### 2.8. FTIR Spectra of Collagen

FTIR spectroscopy of collagen samples was collected using a PerkinElmer MIR/NIR Frontier spectrometer in MIR mode. The data were analyzed using the software program SPECTRUM (PerkinElmer, Hopkinton, MA 01748, USA). The recorded spectra were obtained from a PerkinElmer MIR/NIR Frontier spectrometer, with spectra wavenumbers from 4000 to 400 cm$^{-1}$.

### 2.9. Thermal Stabilization of ASC by DSC (Differential Scanning Calorimeter)

The sample was conducted by dissolving ASC in 0.1 M acetic acid at a solid/acetic acid ratio of 1:40 (*w/v*). The thermal stability was performed by DSC (Pyris 1; PerkinElmer Co., Ltd., Yokohama, Japan) with a rate of scanning of 1 °C/min in the temperature range of 20–50 °C as described by [8]. By estimating the endothermic peak of the DSC

Thermogram, the total denaturation enthalpy ($\Delta H$) and the denaturation temperature ($T_d$) were determined.

### 2.10. Data Analysis

All of the experiments were carried out in triplicate. The data was shown as a standard deviation of the mean (S.D.M). Using Duncan's tests we determined the variable differences, and analyses of data were performed in SPSS 16.0 software.

### 2.11. Animal Welfare

The research was carried out according to the project CT2020.01.TCT.03, approved by the Director of the Department of Science, Technology, and Environment, under the Minister of Education and Training of Vietnam on 9 January 2020.

All experiments were carried out in accordance with national guidelines on the protection of animals and experimental animal welfare in Vietnam following the Law on Animal Health, 2015, Vietnam National Assembly, No. 79/2015/QH13, approved 19 June 2015.

## 3. Results and Discussion

### 3.1. Extraction Yield and Viscosity of ASC According to Snakehead Fish Used Part

The extraction yields and viscosity of ASC from the skin and the mixture of skin and scale of snakehead fish are shown in Table 1. The ASC yield in snakehead fish skin (13.6%) was higher than the yield of ASC from the mixture of skin and scale (12.09%). However, these extraction yields from by-products of snakehead fish (12.09–13.6%) showed a similar tendency of collagen from the skin of tra catfish (13.81%) and tilapia (12.5%), but was lower than the yield of ASC from clown knifefish skin (16.04%) [8] and black carp skin (15.5%) [17]. Thus, Duan et al. [18] reported that the ASC extraction yield was dependent on the fish skin by the differences in skin matrices and the dispensation of alignment in the constituent of skin.

**Table 1.** The extraction yield (%) and viscosity (mPa·s) of ASC from by-products of snakehead fish.

| Sample | Recovery Yield (%) | Viscosity (mPa·s) |
| --- | --- | --- |
| Skin collagen | 13.60 ± 0.13 [a] | 37.5 ± 1.30 [a] |
| Skin and scale collagen | 12.09 ± 0.20 [b] | 21.3 ± 0.95 [b] |

Data are expressed as mean ± standard deviation ($n$ = 3). Different superscripts in the same column indicate statistical differences ($p < 0.05$).

The viscosity of skin collagen was higher than 1.76 times when compared to collagen from the mixture of skin and scale. It might be explained that the skin matrices were loose in comparison to the skin and scale mixture and showed a higher solubility in the solution of acetic acid during the extraction process than the skin and scale mixture. This leads to a higher extraction yield and higher viscosity of ASC from snakehead fish skin.

### 3.2. Effect of Various pH and NaCl Concentrations on ASC Solubility

The solubility alteration of ASC from skin and the mixture of skin and scales at a pH range from 1 to 10 are presented in Figure 1. The high solubility in an acidic pH range, and the solubility decline in the alkaline pHs region by ASC precipitating when pH closed to pI [19], could be observed in both ASCs. Collagen samples showed maximal solubility at pH two. The solubility of both collagen samples showed a slight increase at pH above nine because of the increased repulsion of collagen chains, as the pH gain to a higher value than the pI [20]. This similar tendency in the changing of ASC solubility in this research was agreeable to the reports of [9,18] about the solubility of skin ASC from golden carp and three freshwater fish (tra catfish, clown knife fish, and tilapia), respectively.

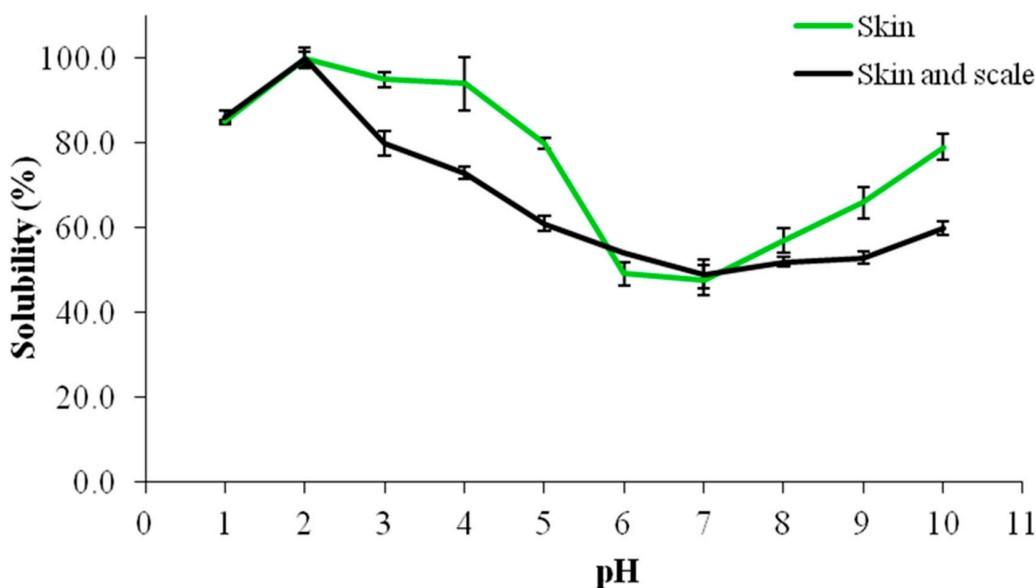

**Figure 1.** The solubility of ASC from by-products of snakehead fish at different pH values.

NaCl concentration also had a large effect on the ASC solubility. Figure 2 showed the solubility of both ASCs at various NaCl concentrations. The similar solubility behaviors could be observed in both collagen samples, with high solubility at the range of NaCl concentrations from 0.2 to 0.4 M (92.16–100%). The dramatic decline in the solubility was shown with NaCl concentrations above 0.4 M because the precipitation of protein by the hydrophobic–hydrophobic interactions was increased [21].

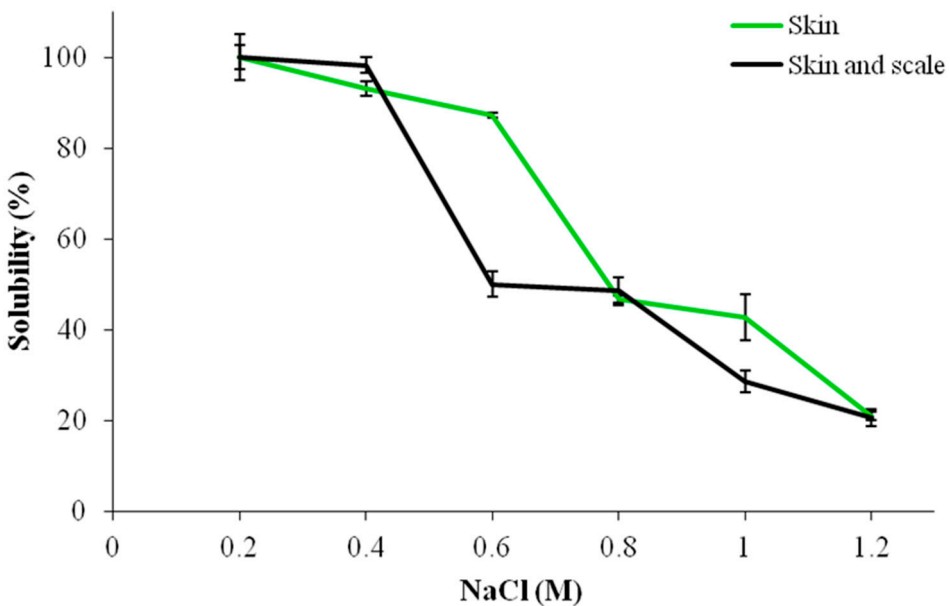

**Figure 2.** The solubility of ASC from by-products of snakehead fish at different NaCl concentrations.

### 3.3. Amino Acid Composition of ASC from Snakehead Fish

Amino acid compositions from the ASC of skin and the mixture of snakehead fish skin and scales are presented in Table 2.

**Table 2.** Amino acid content of collagen from skin and skin-scale of snakehead fish.

| Amino Acid | Amino Acid Content (Residues/1000 Residues) | |
| --- | --- | --- |
| | ASC from Skin | ASC from Skin and Scale |
| Aspartic acid | 54 ± 3 | 49 ± 2 |
| Threonine | 22 ± 2 | 21 ± 2 |
| Serine | 35 ± 2 | 36 ± 3 |
| Glutamic acid | 74 ± 4 | 76 ± 4 |
| Glycine | 307 ± 7 | 314 ± 9 |
| Alanine | 89 ± 5 | 91 ± 6 |
| Valine | 26 ± 1 | 23 ± 1 |
| Cystein | 2 ± 1 | 2 ± 1 |
| Methionine | 12 ± 1 | 12 ± 2 |
| Tryptophan | 0 | 0 |
| Isoleucine | 9 ± 2 | 9 ± 2 |
| Leucine | 28 ± 3 | 24 ± 3 |
| Tyrosine | 5 ± 1 | 4 ± 1 |
| Phenylalanine | 18 ± 2 | 17 ± 2 |
| Hydrolysine | 6 ± 1 | 6 ± 1 |
| Lysine | 31 ± 2 | 30 ± 3 |
| Histidine | 6 ± 1 | 6 ± 1 |
| Arginine | 56 ± 3 | 54 ± 2 |
| Hydroxyproline | 94 ± 5 | 95 ± 4 |
| Proline | 126 ± 5 | 131 ± 6 |
| Imino acid (Proline and Hydroxyproline) | 220 ± 6 | 226 ± 7 |

The amino acid compositions of ASC from snakehead fish by-products, expressed as residues/1000 residues, showed a similar tendency. It is well known that collagen type I is abundant in glycine, which accounted for 307 and 314 residues/1000 residues, from ASC of fish skin and ASC of skin and scale mixture. The $\alpha$-chains in the triple-helix structure of collagen had the general formula Glycine-Proline-Hydroxyproline [22]. The total amount of proline and hydroxyproline (imino acid) of ASC from the skin (220 residues) and the mixture of skin and scale (226 residues) was higher than the ASC from tra catfish, tilapia, and clown knifefish skin (192, 195, and 200/1000 residues, respectively) [8] as well as the skin collagen of Pacific cod (157 residues, [23]) and skin of bighead carp (165 residues, [24]). Normally, the imino acids contribute to the stabilization of collagen via sustaining the wholeness of the triple helix structure. Why the content of imino acid has been different among collagens from various fish species is most likely due to the temperature in the fish habitats [25]. Higher imino acid content of ASC is found in fish species living in warm compared to cold water fishes [10].

### 3.4. Protein Pattern of ASC from Snakehead Fish

SDS-PAGE patterns of collagen from fish skins and the mixture of skin and scale are shown in Figure 3.

SDS-PAGE profiles showed a similar pattern in the protein of ASC from skin and mixture of skin and scale. All ASCs included $\alpha$1, $\alpha$2, as well as $\beta$ chains as major components; the low band intensity of the $\gamma$ chain was found at a very low content, and was identified as a type I collagen. Previous reports have classified ASCs from fish by-products such as skin and scale as collagen type I, including the skin of tilapia, clown knifefish, and Pacific cod [8,23], as well as in the scales of horse mackerel and flying fish [10].

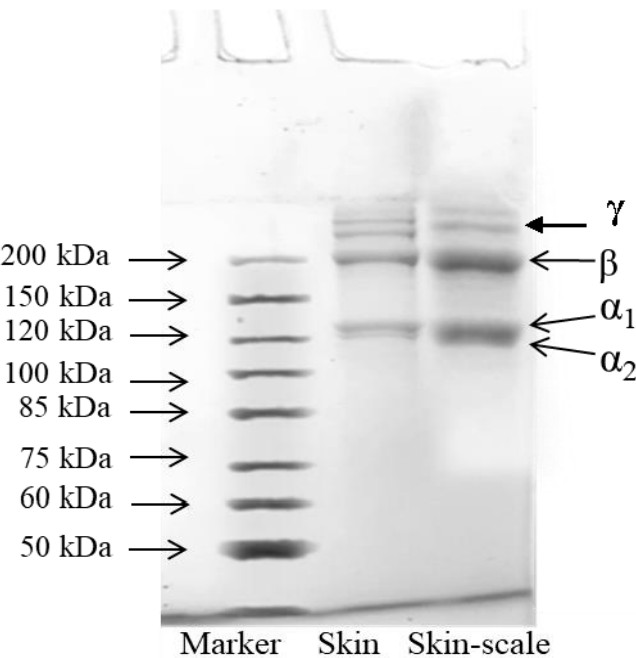

**Figure 3.** Protein patterns of collagen from skin and the mixture of skin and scale of snakehead fish.

*3.5. FTIR Spectra of ASC from Snakehead Fish*

The FTIR spectra of skin and the mixture of skin and scale ASCs from snakehead fish are shown in Figure 4. Both samples had amide A (3400–3440 cm$^{-1}$), amide I (1600 to 1700 cm$^{-1}$), amide II (1500 to 1600 cm$^{-1}$), and amide III (1200 to 1300 cm$^{-1}$) bands, which were in agreement with the research of [9]. Amide A of both ASCs was observed at a wavenumber of 3422 and 3414, respectively. Furthermore, the spectra wavenumber included amide I (1663 and 1642 cm$^{-1}$ in ASC from skin and the mixture of skin and scale, respectively), amide II (1564 and 1552 cm$^{-1}$), and amide III (1206 and 1203 cm$^{-1}$). Amide A, normally associated with N–H stretching vibrations coupled with hydrogen bonds, appears in the spectra range of 3400–3440 cm$^{-1}$. Amide I bands are associated with C=O stretching vibrations in peptides, with the main function to form a secondary protein structure. Amide II (~1500 cm$^{-1}$) represents N–H bending coupled to C–N stretching. The triple-helix structure of collagen is involved with amide III [20]. The absorption bands of amide I and amide III from the skin ASC were higher than the mixture of skin and scale ASC, indicating an association between the transition of α-helix by uncoupling of intermolecular cross-links and the interruption of intramolecular bonding [26]. Moreover, it may lead to the thermal stability of ASC from skin and scale mixture being higher than that from skin collagen. The correlation between denaturation temperature of ASC and the decline of wavenumber of amide I and amide III bands has been described in the study by Thuy et al. [8], who reported that ASC from the skin of clown knifefish, with the lowest wavenumber of amide I and amide III, showed the highest thermal stability in comparison to collagen from tra catfish and tilapia skin.

*3.6. Thermal Properties of ASC from Snakehead Fish*

The temperature of collagen denaturation from snakehead fish skin and the mixture of skin and scale are presented in Figure 5. The endothermic peak of skin ASC was observed with $T_d$ of 34.21 °C, which was slightly lower than that of ASC from the mixture of skin and scale (35.78 °C). Some previous research reported the difference of $T_d$ from different collagen sources and suggested that $T_d$ depended on fish species, fish tissues used for extraction, age, habitat temperature and environments, and seasons [10,27–29]. The denaturation temperature of collagen had a close relationship with the imino acid content [10,12]. Furthermore, Thuy et al. [8] reported that the denaturation temperature of

ASC might not only depend on the imino acid content but is also directly related to the degradation of wavenumber in the amide I and amide III region. ASC from skin and scale mixture with higher content of imino acid (226 residues/1000 residues of amino acid) and a lower wavenumber in amide I and amide III was shown with the $T_d$ higher than that of skin collagen (220 residues).

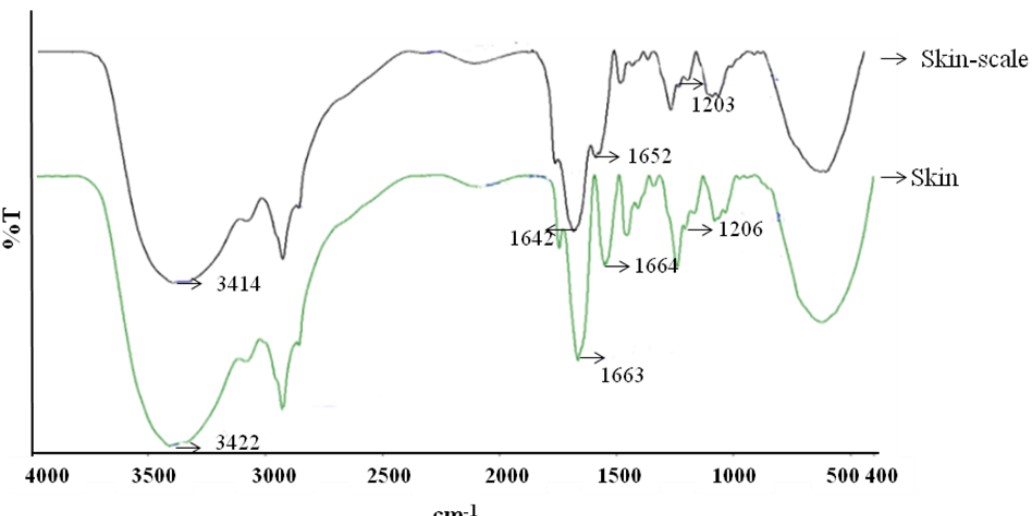

**Figure 4.** FTIR of collagen from skin and skin-scale mixture of snakehead fish.

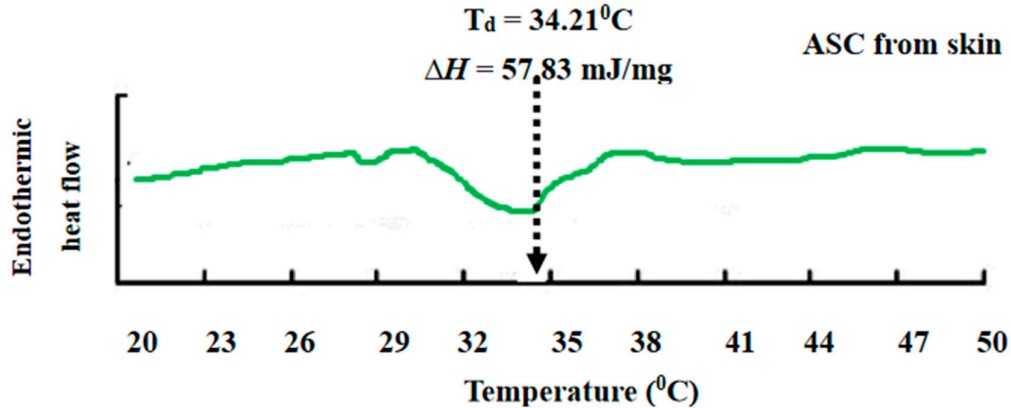

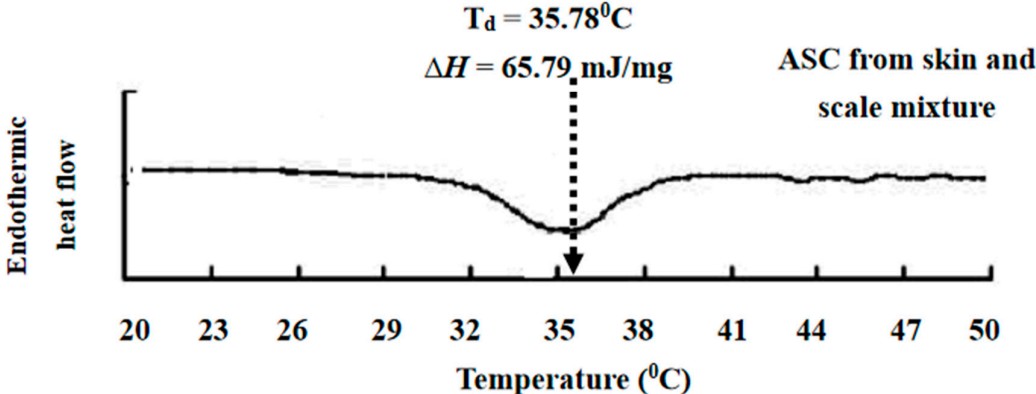

**Figure 5.** Denaturation temperature of collagen from by-products of snakehead fish.

## 4. Conclusions

Based on SDS-PAGE, both ASC from snakehead fish skin and the mixture of skin and scale have been isolated and classified as type I collagen. ASC from fish skin showed the recovery yield was higher than those from the mixture of fish skin and scale. However, collagen samples from skin and scale had higher thermal stability than skin collagen. Thus, the by-products from snakehead fish could be an alternative source for collagen extraction to reduce environmental pollution and increase the value of fish processing by-products. Collagen from snakehead fish by-products could be applied in food science, especially in food packaging.

**Author Contributions:** T.M.T.L., V.M.N., and T.T.T. designed this study and interpreted the results. T.M.T.T. collected test data and drafted the manuscript. All authors have read and agreed to the published version of the manuscript.

**Funding:** This research was funded by Ministry of Education and Training, Vietnam, grant number CT2020.01.TCT.03.

**Institutional Review Board Statement:** The study was conducted according to the national guidelines on the protection of animals and experimental animal welfare in Vietnam following the Law on Animal Health, 2015, Vietnam National Assembly, No. 79/2015/QH13, approved 19 June 2015.

**Informed Consent Statement:** Not applicable.

**Acknowledgments:** This research cost was supported by the CT2020.01 program (CT2020.01.TCT.03 project) funding from the Ministry of Education and Training, Vietnam.

**Conflicts of Interest:** None of the authors have any conflicts of interest.

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
