# Peer review of "Characterization of Acid-Soluble Collagen from Food Processing By-Products of Snakehead Fish (Channa striata)"

_processes, doi:10.3390/pr9071188_

Round 1

Reviewer 1 Report

My comments are the following:

  1. The aim in the abstract is not clearly written.
  2. It should be stated in the abstract the possible application of gained collagen.
  3. The Introduction part is very short. It should be extended at least by the fact how collagen gained in this way can be used. The following reference should be used: Jancikova, S., Jamróz, E., Kulawik, P., Tkaczewska, J., & Dordevic, D. (2019). Furcellaran/gelatin hydrolysate/rosemary extract composite films as active and intelligent packaging materials. International journal of biological macromolecules, 131, 19-28.
  4. Lines 199-217: why they are italic?
  5. The number of references is too small. It should be extended by other references, especially discussion part.

The work is interesting, but the exact application should be written.

Author Response

It is our pleasure to received you review for the manuscript entitled “Characteristic of acid-soluble collagen from by-products of snakehead fish (Channa striata)” for consideration to publish in Processes.  I send to you point-to-point response to your review. Please see the attachment.

Reviewer 2 Report

The manuscript entitled: Characteristic of acid-soluble collagen from by-products of snakehead fish

(Chitala striata) )” describes the isolation of acid-soluble collagen from snakefish by-products. The skin and scales were studied. This article brings no analytical novelty is something that is commonly studied.

Table 1: you pay attention to decimal digits. Also in the text line 166-168.

I had already reviewed this manuscript, and some parts in materials and methods were different, compared to this article by the same authors https://doi.org/10.1016/j.foodchem.2013.10.094.

The authors have now resubmitted a new version of the manuscript by changing the procedure as described in the article I cited above.

Did they get the description wrong the first time?

The corrections I asked for, have been made.

Having clarified this doubt, it can be accepted for me, it is not new, but it is the first application to snake fish and its by-products

Author Response

It is our pleasure to receive your review the manuscript entitled “Characteristic of acid-soluble collagen from by-products of snakehead fish (Channa striata)” for consideration to publish in Processes.  I send to you point-to-point response. Please see the attachment.

Reviewer 3 Report

No novelty in this manuscript.

Author Response

It is our pleasure to receive your review the manuscript entitled “Characteristic of acid-soluble collagen from by-products of snakehead fish (Channa striata)” for consideration to publish in Processes.  

For your comment: No novelty in this manuscript.

Response: 

Thank you very much for your comment. I think the new finding in my research is the utilization snakehead fish by-products with a large amount discarded during their processing in Vietnam for producing the add-value product to increase their value and decrease the environmental pollution. We are on the way to apply collagen from snakehead fish by-products in food packaging.

Kind regards (on the behalf of the authors)

Tran Thanh Truc and Le Thi Minh Thuy

Round 2

Reviewer 1 Report

The manuscript can be accepted.

Author Response

Thank you very much for your acceptance.

Kind regards (on the behalf of the authors)

Tran Thanh Truc and Le Thi Minh Thuy

This manuscript is a resubmission of an earlier submission. The following is a list of the peer review reports and author responses from that submission.

Round 1

Reviewer 1 Report

The novelty in this paper is not clear. It should be clarified why collagen was prepared from a mixture of leather and scales. Establishing a method for preparing collagen for industrial application has great advantages. However, it is meaningless to prepare collagen by the same method as before. If the raw material is new, it is necessary to compare it with other fish species. 

Reviewer 2 Report

The manuscript entitled: “Characteristic of acid-soluble collagen from by-products of snakehead fish (Chitala striata)” describes the isolation of acid-soluble collagen from snakefish by-products. The skin and scales were studied. This article brings no analytical novelty is something that is commonly studied.

Authors should investigate the following points to improve the manuscript:

  • In the cited source https://doi.org/10.1016/j.foodchem.2013.10.094, the contact with the alkaline solution occurred for 6h.

Why did the authors decrease to 4 hours? Is the time enough?

  • In the cited source, the authors changed the NaOH solution after 3 hours and washed completely in cold distilled water until a neutral pH was obtained.

Why didn't you make this step?

  • Paragraph 2.5. Write the equations used to calculate the (%) of solubility.
  • Line 94, What is the oxidant solution used?
  • Paragraph 2.6. Has this method been optimized by anyone already? Cite the authors.
  • Were the samples soluble in the buffer used? Samples are usually solubilized in 0.1 M acetic acid

Reviewer 3 Report

My comments are the following:

  1. The aim is not clearly written in the abstract.
  2. The observation about application findings should be written at the end of the abstract too.
  3. The introduction part is way too short. The reason for the production of collagen should be mentioned too. The following reference should be used: Jancikova, S., Jamróz, E., Kulawik, P., Tkaczewska, J., & Dordevic, D. (2019). Furcellaran/gelatin hydrolysate/rosemary extract composite films as active and intelligent packaging materials. International journal of biological macromolecules131, 19-28.
  4. Line 61: the producer of lyophilisator should be written.
  5. Table 2: there are no standard deviations and statistical analysis
  6. Line 184: the spelling mistake.